# A Robust Adaptive Objective Power Allocation in Cognitive NOMA Networks

**DOI:** 10.3390/s23094279

**Published:** 2023-04-26

**Authors:** Mingyue Zhou, Xingang Guo

**Affiliations:** College of Computer Science and Engineering, Changchun University of Technology, Changchun 130012, China

**Keywords:** cognitive radio, power allocation, spectrum sharing

## Abstract

Cognitive radio (CR) is a candidate for opportunistic spectrum implementation in wireless communications, allowing secondary users (SUs) to share the spectrum with primary users (PUs). In this paper, a robust adaptive target power allocation strategy for cognitive nonorthogonal multiple access (NOMA) networks is proposed, which involves the maximum transmission power of each SU and interference power threshold under PU constraints. By introducing the signal-to-interference-plus-noise ratio (SINR) adjustment factor, the strategy enables single-station communication to achieve energy efficiency (EE) or high throughput (HT), thus making the target function more flexible. In the same communication scenario, different cognitive users can choose different communication targets that meet their needs. Different QoS can be selected by the same cognitive user at different times. In the case of imperfect channel state information (CSI), semi-infinite (SI) constraints with bounded uncertainty sets are transformed into an optimization problem under the worst case, which is solved by the dual decomposition method. Simulation results show that this strategy has good adaptive selectivity and robustness.

## 1. Introduction

The rapid popularization of smart devices is aimed at improving the network capacity and spectrum utilization of wireless networks. Because of the limited availability and low efficiency of the radio spectrum, new perspectives on spectrum use are challenging traditional methods of spectrum management. CR network is a promising solution that can improve the current situation of insufficient spectrum utilization while adapting to the increasing services and applications in wireless networks [1].

Power allocation is an important technology for underlay CRNs. It has mainly been used to reduce co-channel interference and ensure the SINR of SUs, resulting in a better quality of service (QoS). Recent works [2,3,4,5] have focused on maximizing total data throughputs of SUs under perfect CSI. Under the condition of ensuring the maximum tolerance of interference constraints, the time-switching factor and power distribution coefficient are optimized based on statistical CSI in ref. [2]. In ref. [3], a joint optimization of power allocation and NOMA-SU assignment is investigated to achieve maximum throughput in the worst case of all SUs in cognitive NOMA networks. In [4], the problem structure is studied to simplify the maximization of the PU sum rate under the constraint of the minimum SU sum rate; then, it is transformed into a convex problem, and finally, it is solved optimally. In [5], the authors propose a path throughput maximization optimization problem considering QoS and available energy constraints, which are solved by hybrid game theory routing and the power control algorithm. Under the constraints of energy causality and interference power constraints, the end-to-end throughput maximization problem with optimal time and power allocation is studied in [6]. The authors in [7] investigate the power allocation algorithm of the downlink CRN with NOMA, which aims to maximize the number of SUs accessing the CRN-NOMA and the total rates in the system. In [8], the maximization of the sum-date of secondary backscatter sensors (SBSs) is investigated, which is transformed into a convex optimization problem. A particle-swarm-optimization-based power allocation and relay-selection scheme is investigated in [9] by maximizing the sum throughput in a cooperative relaying CR NOMA system.

On the other hand, EE has attracted extensive attention in the research field of wireless communication networks. EE provides new freedom to further improve system performance. A nonlinear fractional programming (NFP) problem with maximum EE is established in [10], for considering SU and antenna height, transmit rate limit, the total power consumption constraint and power consumption on a single subcarrier constraint. A downlink robust resource allocation algorithm in [11] is proposed for robust transmission to maximize the sum energy efficiency of SUs under channel uncertainties. In [12], the authors study the EE issue in 5G communications scenarios, where cognitive femtocells coexist with picocells operating at the same frequency bands. Optimal energy-efficient power allocation based on sensing-based spectrum sharing is proposed for the uplink cognitive femto users operating in a multiuser, multiple input, multiple output (MIMO) mode. The optimal power allocation schemes for a delay-insensitive green CR are proposed in [13] to maximize the EE of the SU. The constraints of peak interference power and average/peak transmitting power are also considered. Paper [14] proposes an EE power control scheme using game in hybrid NOMA-based CRNs.

In practical applications, CSI obtained by SU-TX may be imperfect due to inevitable practical problems such as channel estimation error, outdated information caused by feedback delay, etc. To circumvent this problem, the robust power allocation strategy has been studied to explicitly consider these errors. In [15], the authors study an underlay cognitive NOMA-based coordinated direct and relay transmit network with imperfect CSI. Paper [16] studies the design of a transmit precoder for an SU coexisting with a PU in a MIMO CR, where the CSI at SU-Tx is inaccurate. In [17], a resource allocation model considering an imperfect channel is studied, which can reduce the energy consumption of the PU-TX and improve the adaptability to the uncertain communication environment while ensuring the QoS. In [18], the authors study a robust EE-based maximization resource allocation problem by jointly optimizing power allocation, subcarrier allocation and transmission duration under the consideration of spectrum sensing errors and channel uncertainties simultaneously. Ref. [19] proposes the robust power allocation method to improve robustness against the CSI uncertainties for an underlying cellular network. In [20], a power allocation strategy considering robust EE is studied to appropriately ensure the SINR requirement for SUs and the interferences power threshold at the PU-Rx receivers.

Most of the mentioned studies discussed the resource allocation problems by focusing on one objective, limiting the potential of CRNs. Emerging CRNs need to offer multiple services with distinct characteristics. In addition to traditional voice services, which have strict QoS requirements, some data services, such as www browsing and file downloading, tolerate greater variations in link quality. From the perspective of transmission power control, this mixed-services scenario requires different processing methods, which can take advantage of the change in communication link quality to achieve efficient power allocation.

In this paper, a dynamic adaptive objective power allocation strategy is proposed. The SINR of a communication link can be controlled according to the objective function such as EE and HT. The proposed strategy makes high flexibility the target by taking advantage of SINR variation.

The main contributions of this paper are as follows:

First, our problem representation has not been studied in the traditional CRN. Most previous articles have focused on a target function. In the same communication scenario, different cognitive users can choose different communication targets that meet their needs. Different QoS can be selected by the same cognitive user at different times. The scenarios considered in this article are more challenging. The network flexibility that this scenario brings helps realize the full potential of CR systems.

Second, we investigate the robust power allocation problem for downlink multiuser CRNs to maximize the EE or TH of SUs to the maximum transmit power constraint and interference temperature constraints at PUs. In this framework, we propose the virtual SINR mechanism for dynamic targeting. Compared with the traditional method, it can select the target function more efficiently and satisfy the user’s QoS requirements more appropriately. This power control strategy allows for flexible QoS provision. A distributed power control scheme with fast target QoS adaptation is proposed.

Third, the worst-case method is adopted under imperfect CSI, and channel uncertainty is modeled with a given ellipsoid region, which is transformed into the worst-case optimization problem, namely the maximum-value problem. The results of the simulation show that compared to existing algorithms, the proposed power control algorithm is practical and robust when users have different objectives.

The rest of this paper is organized as follows. Section 2 gives a brief overview of the related work, the system model of the CR system and the formulation of the problem. Section 3 discusses the imperfect CSI case. We propose a worst-case optimization problem and transform the original problem into a semi-infinite programming problem, which can be transformed into a worst-case optimization problem and solved by the Lagrange optimization method. Numerical results for evaluating the performance of the system are provided and discussed in Section 4. Section 5 summarizes this paper.

## 2. System Model and Problem Formulation

The system model is presented, and we describe the formulation of the power control scheme. CRNs allow SUs to dynamically use part of the PU spectrum to communicate without causing harmful interference to the PU. This solves the problem of a scarce spectrum.

We consider an ad hoc CRN with no central control node. The system model considered consists of a PU network and CRNs, as shown in Figure 1. Suppose I cognitive user links are randomly assigned to the cluster area.

The different parameters used in our work are defined in Table 1.

Active transmitter/receiver pairs interfere with each other by simultaneously transmitting over the same channel. Thus, the characteristics of a technical system can be abstracted, since the importance of a distributed power control program is related only to signal and interference power. We represent the transmitted power as $p_{i}$. The SINR received by the SU receiver (SU-RX), i.e.,
(1)$$r_{i}=\frac{g_{ii}p_{i}}{\sum_{j\neq i}^{}g_{ij}p_{j}+g_{i0}p_{0}+n_{i}}$$
where
(2)$$\mu_{i}=\frac{1}{\sum_{j\neq i}^{}G_{ij}p_{j}+N_{i}}$$
is the effective channel gain of CR link i. The collection of all SU transmitters that interfere with link i is represented by M. Where
(3)$$N_{i}=\left(g_{i0}p_{0}+n_{i}\right)/g_{ii}$$
is the effective background noise of CR link i, and
(4)$$G_{ij}=\left\{\begin{array}{c} \begin{array}{ccc} g_{ij}/g_{ii} & , & j\neq i \end{array}\\ \begin{array}{ccc} 0 & , & j=i \end{array} \end{array}\right.$$

In a CRN, each secondary transmitter must respect the total power constraint of each SU to get the QoS of the corresponding link. The power constraint for each SU-TX is
(5)$$p_{i}\leq p_{i}^{max}$$
where $p_{i}^{max}$ is the maximum transmit power of CR-TX i.

It is assumed that all SUs can share the same frequency resource with PUs, subject to the constraint that the total interference generated by SUs does not exceed the interference power threshold tolerated by PU-TX, namely
(6)$$\sum_{i}^{M}p_{i}h_{i}\leq IT$$
where IT is the interference power threshold from a PU.

The SINR error of SU i is defined by
(7)$$e_{i}=r_{i}-r_{i}^{tar}$$
which indicates the deviation between the SU’s target QoS and the received QoS. The adjustment of transmitted power by the SU to minimize SINR deviation is a behavior that is beneficial to the user. The SU’s adjustment of transmitted power in order to reduce the interference introduced by itself should be understood as a behavior beneficial to other network users or networks, as well as energy-efficient behavior.

To dynamically adjust resource allocation, we propose the performance standard of SU i as
(8)$$e_{i}=r_{i}-r_{i}^{tar}$$
which considers both the EE and HT objectives. $\Psi_{i}^{t}$ is the HT adjustment parameter. We call $\left(1+\Psi_{i}^{t}\right)r_{i}^{tar}$ a virtual target SINR. The non-negative $\Psi_{i}^{t}$ in (8) defines the difference in the levels of importance of the EE metric or the HT metric. For different applications, different $\Psi_{i}\left(t\right)$ may be chosen. When $\Psi_{i}^{t}=0$, only the index centered on EE is concerned. Conversely, the HT centric measurement is addressed by choosing the parameter $\Psi_{i}^{t}$ when $\Psi_{i}^{t}>0$.

Dynamic adjust resource allocation is formulated as
$$min{\left(r_{i}-\left(1+\Psi_{i}^{t}\right)r_{i}^{tar}\right)}^{2}$$
(9)$$s.t.\left\{\begin{array}{c} C1:p_{i}\leq p_{i}^{max}\\ C2:\sum_{i}^{I}p_{i}h_{i}\leq IT \end{array}\right.$$

Equation (9) is a nonlinear programming problem; the Lagrange algorithm can be used to find the optimal transmit power. This strategy does not consider the disturbance of channel gains and is called a non-robust algorithm (NRA).

## 3. Robust Energy Efficiency Problem Formulation

Both SUs and spectrum holes can come and go, making cognitive radio networks a highly dynamic and challenging wireless environment [21]. Therefore, it is very important to find a robust resource allocation algorithm. It can implement reasonably good solutions fast enough to guarantee acceptable performance levels under worst-case interference conditions.

We consider the imperfect CSI in (9) and describe the uncertainty using a set of ellipsoids. The robust power control problem can be expressed as a worst-case optimization problem.

The channel gain between CR-TX i and PU-RX is expressed as
(10)$$h_{i}=\tilde{h_{i}}+\Delta h_{i}$$
where $\tilde{h_{i}}$ is the nominal part. $\Delta h_{i}$ is the disturbance value. Suppose H is the uncertain set of vectors h. An ellipsoid is used to depict the uncertain H, i.e.,
(11)$$h_{i}=\tilde{h_{i}}+\Delta h_{i}$$
where $\xi^{2}$ is the maximum deviation of each item in h.

Let $G_{i}$ note the channel uncertainty set of ith row for the matrix G. $G_{i}$ describes the disturbance interference with the channel gain associated with the channel gain for link i. An ellipsoid is to describe the set $G_{i}$. The normalized channel gain from CR-TX j to CR-Rx i is expressed as
(12)$$G_{ij}=\tilde{G_{ij}}+∆G_{ij}$$
where $\tilde{G_{ij}}$ is the nominal part, and $∆G_{ij}$ is the disturbance value. The ith line of $\tilde{G}$ is called $\tilde{G_{i}}$, and the corresponding perturbation is called $∆G_{i}$. In the approximation of an ellipsoid, the uncertainty set of $G_{i}$ for $G_{i}$ can be described as
(13)$$G_{i}=\left\{\tilde{G_{i}}+\Delta G_{i}:\sum_{j\neq i}^{}{\left|∆G_{ij}\right|}^{2}\leq \zeta_{i}^{2}\right\}$$
where $\zeta_{i}$ is the maximum deviation of each item in $G_{i}$.

The robust power control problem considering channel uncertainty can be expressed as
$$min{\left(\mu_{i}^{\prime }p_{i}-\left(\left(1+\Psi_{i}^{t}\right)r_{i}^{tar}\right)\right)}^{2}$$
(14)$$s.t.\left\{\begin{array}{c} C1:p_{i}\leq p_{i}^{max}\\ \overset{-}{C2}:\sum_{i}^{I}\tilde{{(h}_{i}}+\Delta h_{i})p_{i}\leq IT\\ C3:\sum_{i}^{}{\left|∆h_{i}\right|}^{2}\leq \xi^{2}\\ C4:\sum_{j\neq i}^{}{\left|∆G_{ij}\right|}^{2}\leq \zeta_{i}^{2} \end{array}\right.$$
where
(15)$$\mu_{i}^{\prime }=\frac{1}{\sum_{j\neq i}^{}\tilde{{(G}_{ij}}+∆G_{ij})p_{j}+N_{i}}$$
is the robust effective channel gain of CR link i. Obviously, Equation (14) is a SIP problem, which is converted to an equivalence problem by considering the worst case.

We can use Cauchy–Schwarz inequality which can be applied in many contexts. Cauchy–Schwarz inequality says that if x and y are members of a real or complex inner product space, then
(16)$$\left|<x,y>\right|\leq \left\|x\right\|\left\|y\right\|$$
this is true if and only if x and y are linearly dependent.

For the Cauchy–Schwarz inequality, we have
(17)$$\sum_{i}^{I}\left(h_{i}-\tilde{h_{i}}\right)p_{i}\leq \xi \sqrt{\sum_{i}^{}p_{i}^{2}}$$
(18)$$\sum_{j\neq i}^{}\left(G_{ij}-\tilde{G_{ij}}\right)p_{i}\leq \zeta_{i}\sqrt{\sum_{j\neq i}^{}p_{j}^{2}}$$

In the worst case, the robust effective channel gain and constraint $\overset{-}{C2}$ can be transformed into the following form:(19)$$\sum_{i}^{I}\tilde{h_{i}}p_{i}+\sum_{i}^{I}\Delta h_{i}p_{i}=\sum_{i}^{I}\tilde{h_{i}}p_{i}+\xi \sqrt{\sum_{i}^{}p_{i}^{2}}\leq IT$$
(20)$$\mu_{i}^{\prime }=\frac{1}{\sum_{j\neq i}^{}\tilde{G_{ij}}p_{j}+\sum_{j\neq i}^{}∆G_{ij}p_{j}+N_{i}}\leq \frac{1}{\sum_{j\neq i}^{}\tilde{G_{ij}}p_{j}+\zeta_{i}\sqrt{\sum_{j\neq i}^{}p_{j}^{2}}+N_{i}}=\tilde{\mu_{i}^{\prime }}$$
where $\tilde{\mu_{i}^{\prime }}$ is the worst-case effective channel gain.

The robust optimization problem can be converted to an equivalent problem with only linear constraints and solved by the distributed decomposition theory.

In the worst case, we convert problem (14) to the following nonlinear programming form: $$min{\left(\tilde{\mu_{i}^{\prime }}p_{i}-\left(\left(1+\Psi_{i}^{t}\right)r_{i}^{tar}\right)\right)}^{2}$$
(21)$$s.t.\left\{\begin{array}{c} C1:p_{i}\leq p_{i}^{max}\\ \tilde{C2}:\sum_{i}^{I}\tilde{h_{i}}p_{i}+\xi \sqrt{\sum_{i}^{}p_{i}^{2}}\leq IT \end{array}\right.$$
$P={\left\{p_{1},p_{2},\ldots ,p_{I},\right\}}^{T}$ is the optimization variable in this strategy. Equation (21) is a conservative protection scheme for PU-RX since CSI cannot always be worst-case.

Obviously, Equation (21) is a convex optimization problem which can be computed by the dual decomposition method [20].

The Lagrange function for optimization Equation (21) is as follows:(22)$$L\left(p_{i},\theta_{i},\vartheta_{i}\right)={\left(\tilde{\mu_{i}^{\prime }}p_{i}-\left(\left(1+\Psi_{i}^{t}\right)r_{i}^{tar}\right)\right)}^{2}+\theta_{i}\left(p_{i}-p_{i}^{max}\right)+\vartheta_{i}(\sum_{i}^{I}\tilde{h_{i}}p_{i}+\xi \sqrt{\sum_{i}^{}p_{i}^{2}}-IT)$$
where $\theta ={\left(\theta_{1},\theta_{2},\ldots ,\theta_{I}\right)}^{T}$ and $\vartheta ={\left(\vartheta_{1},\vartheta_{2},\ldots \vartheta_{I}\right)}^{T}$. $\theta_{i}$ and $\vartheta_{i}$ are the Lagrange multiplier. The values of $\theta_{i}$ and $\vartheta_{i}$ are non-negative, and they take the following form:(23)$$\theta_{i}^{t+1}=max\left[\theta_{i}^{t}+\alpha_{1}\left(p_{i}-p_{i}^{max}\right),0\right]$$
(24)$$\vartheta_{i}^{t+1}=max\left[\vartheta_{i}^{t}+\alpha_{2}(\sum_{i}^{I}\tilde{h_{i}}p_{i}+\xi \sqrt{\sum_{i}^{}p_{i}^{2}}-IT),0\right]$$
where $\alpha_{1}$ and $\alpha_{2}$ are the iterative steps of the Lagrange multiplier algorithm.

Under the Karush–Kuhn–Tucker (KKT) conditions, each user can independently calculate the robust optimal transmitting power by the following formula:(25)$$\frac{\partial L\left(p_{i},\theta_{i},\vartheta_{i}\right)}{\partial p_{i}}=0$$

We can get the optimal equilibrium as follows:(26)$$p_{i}^{opt}=max[\frac{\left(1+\Psi_{i}^{t}\right)r_{i}^{tar}}{\tilde{\mu_{i}^{\prime }}}-\frac{\theta_{i}+\vartheta_{i}\tilde{h_{i}}}{2{\tilde{\mu_{i}^{\prime }}}^{2}},0]$$

This is a robust distributed power control algorithm, which guarantees convergence to the optimal value if the step size is small enough. The proposed robust algorithm (RA) can adaptively select the objective function to achieve a certain SINR, where we assume that there is no perfect CSI.

## 4. Computer Simulations and Discussion

In this section, we evaluate the proposed robust adaptive objective algorithm and its ability to provide flexible QoS and efficient power allocation. The performance of the robust power allocation strategy is evaluated for the cognitive NOMA network by numerical examples. There is a primary network and an underlying NOMA network. The SUs are randomly distributed around the PU. Similar to [21], the simulation parameters are shown in Table 2.

Some of the channel parameters in this paper are as follows:(27)$$h_{i}=\left[0.15\;0.11\;0.13\;0.12\;0.13\right],i=1,2,\ldots ,5$$
*g_ij_* = [1.0000 0.0805 0.0523 0.0305 0.1065;    0.1098 1.0000 0.0491 0.0239 0.0902;    0.1095 0.0273 1.0000 0.0645 0.0563;    0.0881 0.0384 0.0297 1.0000 0.0843;    0.0237 0.1072 0.0834 0.0292 1.0000](28)

The background noise is
*n_i_* = [0.0001 0.0012 0.0022 0.0003 0.0011](29)

Figure 2 shows the transmit power of each SU for the NRA scheme with perfect CSI. The SINR parameter adjustment factor is $\Psi_{i}^{t}=0$. This means that the objective function is $min{\left(r_{i}-r_{i}^{tar}\right)}^{2}$. Our goal is to minimize the sum of squares of the difference between the received SINR and the target SINR, which is a power allocation scheme of EE. The transmit power of each SU is observed to be in the range of 0.32 mW to 0.35 mW. The EE NRA mechanism can reach equilibrium rapidly. If we want to get higher throughput, we just need to adjust parameter $\Psi_{i}^{t}$. The larger $\Psi_{i}^{t}$ is, the higher the throughput will be. Next, the simulation is discussed. The magnitude of $\Psi_{i}^{t}$ is related to the interference power threshold and the maximum transmit power.

Channel gain disturbance parameters ξ and $\zeta_{i}$ are both 0.02. All SUs update their transmit power in an EE manner with $\Psi_{i}^{t}=0$. Figure 3a shows the transmit power of each SU for the RA scheme. The transmit power of each SU achieved by the RA scheme can be found to be greater than that achieved by the NRA scheme. It is evident that the power values for the RA scheme lie between 0.36 mW and 0.41 mW because each SU requires higher transmit power to overcome channel uncertainty. Figure 3b shows the SINR for each SU with the RA scheme. The SINRs received by each SU all reach the target SINR. There will be an interrupting event when an SU’s SINR is less than the target SINR value. Figure 3b shows that the RA scheme can guarantee SINR values. In other words, there are no interrupting events that often occur in the NRA scheme because the RA scheme considers the worst-case channel fluctuations. Taken together with Figure 2 and Figure 3, it can be observed that the RA scheme has good robustness and can offer better protection for each SU.

Note that the objective function of the NRA and RA schemes can be varied at any time. The perturbations ξ and $\zeta_{i}$ are both 0, and the initial objective function is the EE manner with $\Psi_{i}^{t}=0$. Clearly, the RA scheme relaxes to the NRA scheme. Each SU transmits power this way. If the number of iterations is equal to 12, our objective function is changed to the HT manner with $\Psi_{i}^{t}=0.05$, for all i. Figure 4 shows the transmit power and SINR for the RA for this situation. Figure 4a shows each SU’s transmit power for the NRA. With perfect CSI, the SUs can obtain more transmit power to guarantee HT for the second time slot, because a bigger $\Psi_{i}^{t}$ means that the high-throughput manner is the dominant approach. Figure 4b presents the curves of convergence of the achieved SINR at the receiver to the corresponding target SINR for a terminal in the cognitive NOMA network. In the EE metric manner, each SU’s SINR can achieve the target SINR value. Then, each one maintains the target SINR for the first time slots. As the target requirement changes to the HT fashion, a high SINR is immediately achieved with a higher transmit power.

We depict the transmit power and SINR versus the different SINR parameter adjustment factors in Figure 5. Figure 5a shows the transmit power for each SU versus $\Psi_{i}^{t}$ subject to ξ=0.02 and $\zeta_{i}=0.02$. During the first time slot, $\Psi_{i}^{t}=0$. When the iteration is equal to 12, $\Psi_{i}^{t}=0.05$. The result shows that the transmit power of the HT manner ($\Psi_{i}^{t}=0.05$) is higher than that of the EE manner ($\Psi_{i}^{t}=0)$. It is noticed that the transmit power under the RA scheme is larger than that of the NRA scheme with the same $\Psi_{i}^{t}$. The reason is that the RA scheme has good robustness to present better protection for the SU as the cost of larger energy. Figure 5b presents SINRs for all SUs. The received SINR of SU-RX i for the HT manner is larger than the EE manner since the task of the EE manner is to adjust the transmit power to the minimum value so that the receiver of the SU can reach the desired SINR level, which is a target SINR tracking power control algorithm. In fact, our algorithm is more suitable for actual communication scenarios because SUs can choose their own objective function at any time. It is impossible for all SUs to pursue EE or HT at the same time. For example, some SUs can choose the EE metric mode, while others can choose the high-throughput metric mode. During the second period, SU2 and SU3 can also choose the EE metric manner. It means that the transmit power and SINR for SU2 and SU3 are the same for the first time slot.

Figure 6 shows the received transmit power and SINR for different $\Psi_{i}^{t}$ with the same channel disturbance. For comparison, the curves of $\Psi_{i}^{t}$ increase, and transmit power and the SINR are also simulated in Figure 6a. From the figure, the greater $\Psi_{i}^{t}$ is, the greater the transmit power will be. It is found that the values of power increase with the $\Psi_{i}^{t}$ increase since a big $\Psi_{i}^{t}$ factor enlarges the upper bound of the virtual SINR. The SINRs for the RA scheme are depicted in Figure 6b. As the $\Psi_{i}^{t}$ increases, the received SINRs for each SU are increased to ensure the high throughput. Furthermore, there is a big gap between different $\Psi_{i}^{t}$ for each SU, which demonstrates that the RA algorithm with high-SINR-parameter technology can greatly improve the system throughput.

## 5. Conclusions

A robust power allocation scheme for an adaptive target function is proposed. In this scenario, each cognitive user adjusts its power in an EE or HT manner mode, respectively, based on a virtual SINR function. Every cognitive user on the network can switch to a goal that meets their communication needs at any time. In order to better understand the design of an adaptive target function and its robustness, the scheme is implemented in perfect CSI and worst-case scenarios considering interference power threshold and maximum transmission power constraints, respectively. An iterative robust multiple-objective power allocation scheme is proposed to transform the SIP problem into a worst-case optimization problem. Simulation results verify the validity and effectiveness of this scheme. Another important feature of the scheme is that the algorithm has strong robustness to channel disturbances and is more suitable for practical application.

In future studies, we will focus on vehicles or SUs that face different communication needs when operating at high speeds. The cooperative transmission of information and energy in relay and multicast scenarios will also be studied. The joint optimization of multi-dimensional resource allocation strategies such as frequency allocation, time, power and moving path is studied in detail.

## Figures and Tables

**Figure 1 sensors-23-04279-f001:**
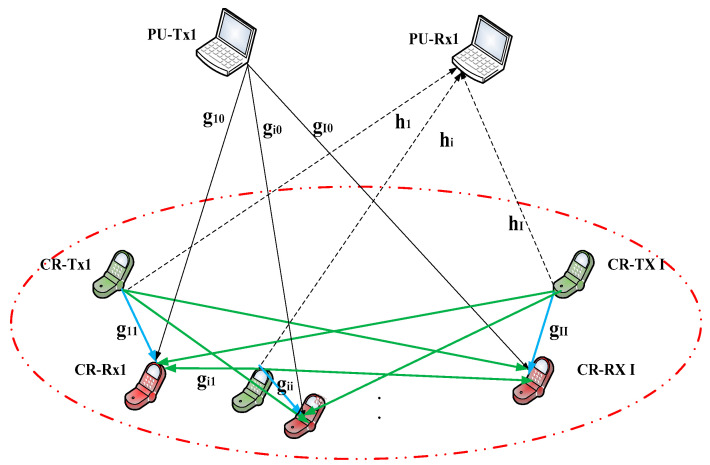
CRN model.

**Figure 2 sensors-23-04279-f002:**
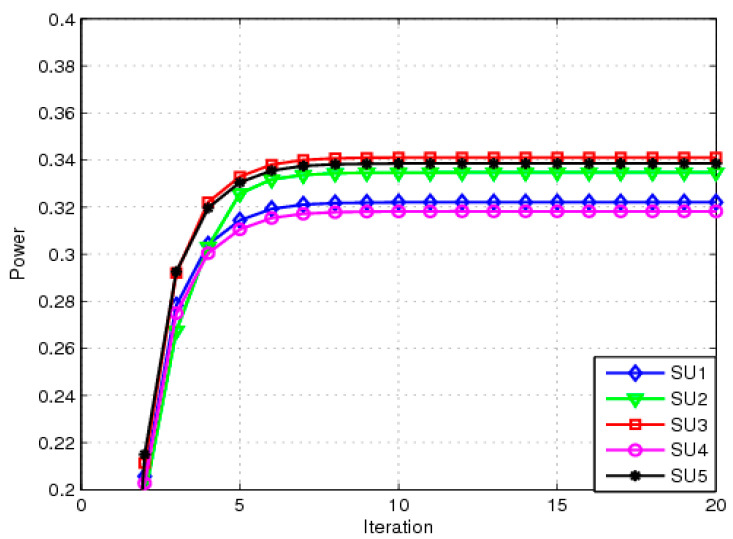
Transmit power for EE NRA.

**Figure 3 sensors-23-04279-f003:**
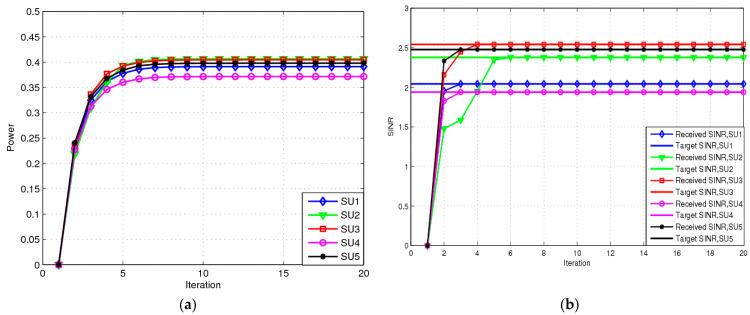
Transmit power and SINA for RA: (**a**) transmit power for RA; (**b**) SINR for RA.

**Figure 4 sensors-23-04279-f004:**
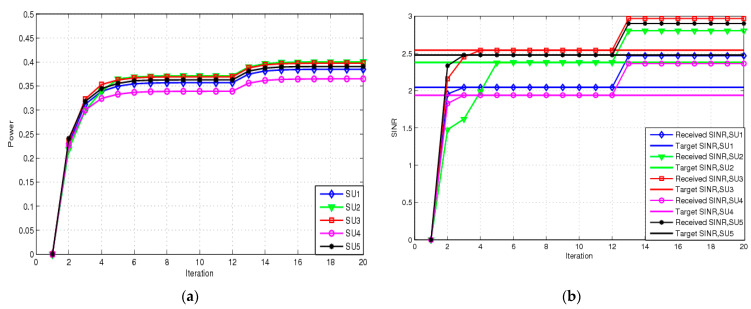
Transmit power and SINA for NRA: (**a**) transmit power for NRA; (**b**) SINR for NRA.

**Figure 5 sensors-23-04279-f005:**
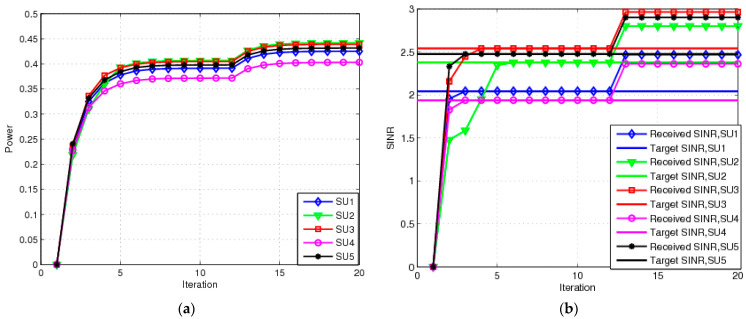
Transmit power and SINR with RA for different $\Psi_{i}^{t}$: (**a**) transmit power for RA; (**b**) SINR for RA.

**Figure 6 sensors-23-04279-f006:**
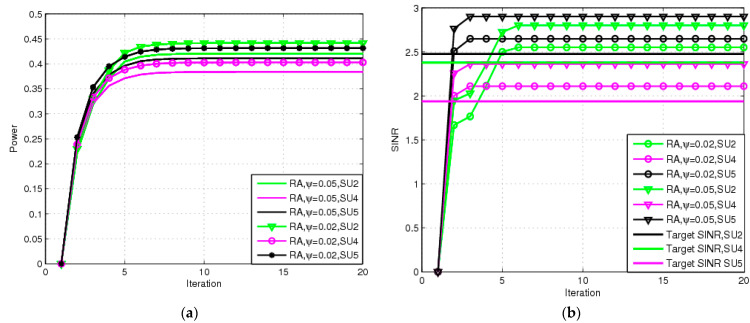
Transmit power and SINR for RA with ξ=0.02 and $\zeta_{i}=0.02$: (**a**) transmit power for RA; (**b**) SINR for RA.

**Table 1 sensors-23-04279-t001:** The list of different symbols and their definition.

Symbol	Meaning
$r_{i}$	the SINR of SU i
$r_{i}^{tar}$	the target SINR of SU i
$\Psi_{i}^{t}$	the time-varying factor of SINR
$p_{i}$	transmit power to SU i
$p_{i}^{max}$	the maximum transmitting power of SU i.
${IT}_{}$	interference power threshold
$g_{ii}$	the channel gain between the CR-TX and the CR-RX of link i
$g_{ij}$	the channel gain from the CR-TX j to the CR-RX i
$g_{i0}$	the interference channel gain from the PU-TX to the CR-RX i
$P_{0}$	the transmit power of PU
$h_{i}$	the channel gain between the CR-TX i and the PU-RX
$n_{i}$	the background noise at the CR-RX of link i

**Table 2 sensors-23-04279-t002:** Simulation parameters.

Parameters	Values
Number of SUs I	5
Number of PUs	1
Background noise $n_{i}$	$\left[0,0.004\right]$ mW
Maximum transmit power $p_{i}^{max}$	$\left[0.82,0.90\right]$ mW
Interference power threshold IT	0.5 mW
Target SINR $r_{i}^{tar}$	$\left[1.9,2.6\right]$ dB
Channel gains $g_{ij}$	$\left[0,1\right]$
Channel gains $h_{i}$	$\left[0,0.2\right]$
Channel gains $g_{i0}$	$\left[0,0.2\right]$
Transmit power for PU	1 mW

## Data Availability

Not applicable.

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
