# Peer review of "A Robust Adaptive Objective Power Allocation in Cognitive NOMA Networks"

_sensors, 2023, doi:10.3390/s23094279_

Round 1

Reviewer 1 Report

Abstract and introduction part needs improvement. Conclusion part is not clear

Reviewer 2 Report

This authors propose a robust adaptive objective power allocation strategy in cognitive non- orthogonal multiple access (NOMA) networks. A signal to interference plus noise ratio (SINR) adjustment factor which enables SUs to meet the communication goals of energy efficiency (EE) or high throughput (HT) is introduced.  A worst-case optimization problem is solved by dual decomposition method. The presented simulation results show the efficiency of the proposed scheme. Although the manuscript contains some interesting research results, the presentation should be improved.  

The selection of the values for all the parameters used in the simulation should be explained.

The analysis of the simulation results should be more elaborate and the conclusions should be more clearly justified.

The quality of the figures should be improved.

English should be checked. There are too many grammatical errors in the text.

Author Response

Please see the attachment。

Reviewer 3 Report

It is suggested that in section 5. Conclusions the results obtained in section 4. Computer Simulations and Discussion be presented in quantitative form. In addition, it is necessary to incorporate in point 5 future works.

Reviewer 4 Report

This paper presents a power allocation scheme for cognitive NOMA networks in order to improve energy efficiency or transmission throughput for secondary users. The authors assume imperfect CSI for analysis and present an optimization problem as a semi-infinite programming problem. The results show that the proposed power control algorithm can achieve the goal of energy efficiency or throughput.

Although the paper attempts to fill the research gap of cognitive NOMA networks, the authors did not describe the challenges of deploying cognitive NOMA networks. For example, the system model does not mention the characteristics of NOMA. Moreover, the mobility of SUs is not considered in the simulation. The authors should tightly couple their scheme with NOMA. The simulation results can also be improved by considering some related schemes. It is unclear whether the proposed scheme is effective when the current manuscript merely shows the performance of different SUs. 

Author Response

Dear reviewer,

Thank you very much for your comments on our manuscript entitled ‘A robust adaptive objective power allocation in Cognitive NOMA Networks’. These comments are valuable, constructive and helpful for the revision and improvement of our paper, as well as significant and important for our future research. We have carefully studied the comments and made modifications and corrections based on your critiques and idea. All the revised parts are marked in blue font in the revised manuscript.

We appreciate for your warm help and all work you have done for us.

The main explanations and the responses to your comments are as the attachment.
